# Comparative Analysis of Phytochemical Profiles and Antioxidant Activities between Sweet and Sour Wampee (*Clausena lansium*) Fruits

**DOI:** 10.3390/foods11091230

**Published:** 2022-04-25

**Authors:** Xiaoxiao Chang, Yutong Ye, Jianping Pan, Zhixiong Lin, Jishui Qiu, Cheng Peng, Xinbo Guo, Yusheng Lu

**Affiliations:** 1Institute of Fruit Tree Research, Guangdong Academy of Agricultural Sciences, Key Laboratory of South Subtropical Fruit Biology and Genetic Resource Utilization, Ministry of Agriculture and Rural Affairs, Guangdong Provincial Key Laboratory of Tropical and Subtropical Fruit Tree Research, Guangzhou 510640, China; xxchang6@163.com (X.C.); panjianping_2003@163.com (J.P.); lzxf200@126.com (Z.L.); rgxoii307@126.com (J.Q.); pengcheng2007@foxmail.com (C.P.); 2School of Food Science and Engineering, South China University of Technology, Guangzhou 510640, China; yytaw95@mail.ubc.ca

**Keywords:** wampee fruit, titratable acid, phenolics, flavonoids, antioxidant activities

## Abstract

As a local medicine and food, wampee fruit, with abundant bioactive compounds, is loved by local residents in Southern China. Titratable acid (TA), total sugar (TS), and total phenolic and flavonoid contents were detected, and phytochemical profiles and cellular antioxidant activities were analyzed by the HPLC and CAA (cellular antioxidant activity) assay in five sweet wampee varieties and five sour wampee varieties. Results showed that the average TS/TA ratio of sweet wampee varieties was 29 times higher than sour wampee varieties, while TA content was 19 times lower than sour wampee varieties. There were much lower levels of total phenolics, flavonoids, and antioxidant activities in sweet wampee varieties than those in sour wampee varieties. Eight phytochemicals were detected in sour wampee varieties, including syringin, rutin, benzoic acid, 2-methoxycinnamic acid, kaempferol, hesperetin, nobiletin, and tangeretin, while just four of them were detected in sweet wampee varieties. Syringin was the only one that was detected in all the sour wampee varieties and was not detected in all sweet wampee varieties. Correlation analysis showed significant positive correlations between TA with phenolics, flavonoids, and total and cellular (PBS wash) antioxidant activities, while there were significant negative correlations between TS/TA with phenolic and cellular (no PBS wash) antioxidant activities. This suggested that the content of titratable acid in wampee fruit might have some relationship with the contents of phenolics and flavonoids. Sour wampee varieties should be paid much attention by breeders for their high phytochemical contents and antioxidant activities for cultivating germplasms with high health care efficacy.

## 1. Introduction

*Clausena lansium* (Lour.) Skeels, commonly known as wampee, is indigenous to and commonly cultivated in Southern China—such as in the provinces of Guangdong, Fujian, Hainan, Guangxi, and occasionally in Sri Lanka, Australia, India, America, and South Asia [1]. Because wampee leaves, stems, and seeds contain functional compounds such as clausenamide, carbazole alkaloids, and essential oils, they are used as traditional medicines and food ingredients for cough [2], viral hepatitis [3], and dermatological and gastro-intestinal diseases [4] due to the following properties: antioxidant [5], antidiabetic [6,7], antimicrobial [8,9], anti-inflammatory [10], hepatoprotective [11], anticancer [12], and nootropic and cerebral protective [13,14], etc. The wampee fruits have a pleasant flavor and are consumed either fresh or served with meat dishes and in preserves. Although less studies have focused on wampee fruits, they were reported containing high phytochemicals (including phenolics and flavonoids), and the extracts from fruits show high antioxidant, anti-inflammation, and anticancer activities [10,15,16].

Epidemiological studies have highlighted that a phytochemical-rich diet protects against chronic diseases [17]. Phytochemicals, such as phenolic acids, flavonoids, anthocyanidins, and tannins, are rich in plant foods [18], which possess remarkable antioxidant and anticancer activities [19], and they can protect against chronic diseases such as cardiovascular diseases, cancers, diabetes, and neurodegenerative diseases [17,18]. Phenolic compounds in fruits have been reported as being positively related to antioxidant activity and have potential health benefits [20,21]. Superfruits contain more bioactive compounds and are consumed regionally; they are gaining popularity in the marketplace due to their nutritional and therapeutic values, including acai, acerola, camu-camu, goji berry, and jaboticaba, among others [22]. Wampee fruits are typically tropical and subtropical fruits and are consumed in Southern Asia as traditional and folk fruits and medicine, which are rich in polyphenols and exhibit high antioxidant activities, as reported [23].

The phenolics and antioxidant activities of different varieties of wampee fruit were analyzed in recent years by our research team, and results showed that the sweet wampee varieties CCTHP and THP, with high soluble solids content and little content of titratable acid, contained less flavonoids and phenolics and had lower antioxidant activities than the sour wampee varieties, YSDH and JFHP [23,24]. Further, we analyzed the phenolic and flavonoid contents and antioxidant activities of wampee fruits of more than one hundred and fifty different germplasms, and we found that sweet wampee varieties contained less phenolics and flavonoids and showed lower antioxidant activities than those of the sour wampee varieties. The hypothesis was that the titratable acid and/or sugar contents might have some relationship with phytochemical compounds such as phenolics and flavonoids in wampee fruits. In order to study this phenomenon, five sweet wampee varieties and five sour wampee varieties were selected in this study, and their titratable acid, total sugar, phenolic and flavonoid contents, and antioxidant activities were analyzed. This work will provide an insight into the relationship between sugars, acids, and phenolic compounds in wampee fruit.

## 2. Materials and Methods

### 2.1. Sample Preparation

The fruits of ten varieties of wampee (*Clausena lansium* (Lour.) Skeels) were collected from the wampee resources nursery in the Institute of Fruit Tree Research, Guangdong Academy of Agricultural Sciences, Guangzhou, China. The cultivars were five sweet wampee varieties (**THP**: TianHuangPi; **ZFHP**: ZaoFengHuangPi; **CCTP**: CongChengTianPi; **LTDH**: LuTianDuHe; **TXTP**: TaXiaTianPi) and five sour wampee varieties (**M3H**: Min3Hao; **JZP**: JiZiPi; **M4H**: Min4Hao; **TXSP**: TaXiaSuanPi; **JXHP**: JiXinHuangPi), as shown in Figure 1. The fruits were harvested freshly in the fully matured stage. Fifty fruits of each variety, without pests and diseases, were selected for handling. The seeds were removed from the fruit, and the residues (pulp and peel) were stored at −20 °C until analysis.

### 2.2. Determination of Titratable Acid and Total Sugar

Total sugar was determined by the anthrone method [20]. Total sugar was determined from the standard curve prepared using glucose and was expressed as “g/100 g FW”. Titratable acid was measured according to the AOAC 962.12 method (AOAC, 2012) with an Automatic Potentiometric Titrator (TITRALAB TIM840, Loveland, Colorado, USA) and was expressed as “g/100 g FW”.

### 2.3. Phytochemical Extraction

Phytochemical contents of the wampee fruits were extracted by the following method, as reported earlier [20,23]. Briefly, 50 g of wampee fruit was homogenized with 400 mL of 80% cold acetone for 3 min. The homogenate was extracted stationary overnight, followed by filtration under reduced pressure. The filtrate was evaporated using a rotary evaporator at 45 °C and redissolved by 70% methanol. All the extracts were stored at −20 °C for the following analysis.

### 2.4. Determination of Total Phenolic and Flavonoid Content

The total phenolics of the wampee fruit extracts were determined by using the Folin-Ciocalteu method [25], and gallic acid was used as the standard for calculation. Total phenolic content was expressed as milligrams of gallic acid equivalents per 100 g of fresh weight (mg GAE/100 g FW). The total flavonoids of wampee fruit extracts were tested by the sodium borohydride/chloranil colorimetric method [26], and catechin was used as the standard for calculation. Total flavonoid content was expressed as milligrams of catechin equivalents per 100 g of fresh weight (mg CE/100 g FW). All the data were reported as mean ± SD for three replicates.

### 2.5. Determination of Phytochemical Profiles

Phytochemical profiles were determined on a Waters HPLC system (Waters Corp., Milford, MA, USA), consisting of a binary pump (model 1525), a micro degasser, an autosampler (model 2707), a thermostatically-controlled column apartment (model 1500), and a photodiode array detector (model 2998). Sample separation was employed with a gradient elution program at the flow rate of 1 mL/min and the column temperature of 30 °C in a Waters HSS T3 C18 column (150 mm × 4.6 mm, 5 μm). The chromatographic data were recorded and processed by Waters software. The mobile phase consisted of 0.1% trifluoroacetic acid solution (aqueous) (A) and acetonitrile (B) using a gradient elution of 10% B at 0–2 min, 10–25% B at 2–7 min, 25–30% B at 7–15 min, 30–58% B at 15–17 min, 58–100% B at 17–18 min, 100% B at 18–19 min, and 100–10% B at 19–20 min. The flow rate of the mobile phase was kept at 1 mL/min. The UV absorbance at 280 nm and 370 nm was monitored for phenolic acids and flavonoids, respectively. Chromatographic peaks were identified by comparing the retention times in specific UV spectra with those of authentic standards. Data were reported as mean ± SD (*n* = 3).

### 2.6. Determination of Antioxidant Activities

The total antioxidant activity of samples was determined using the oxygen radical absorbance capacity (ORAC) assay [27]. Fluorescein disodium salt was used as the fluorescence probe, and 2,2′-azobis (2-amidinopropane) dihydrochloride (ABAP) was used as the free radical donor in this assay. The total antioxidant activity value was calculated by standard Trolox, and the data were expressed as mean ± SD millimole of the Trolox equivalents (TE) per 100 g in fresh weight (mmol TE/100 g FW) for three replicates.

The cellular antioxidant activity (CAA) assay was applied in this study to determine the cellular antioxidant ability of the wampee fruit samples [20,28]. Human live cancer cell line HepG2 (ATCC HB-8065) was used as the cellular model in this assay; quercetin was used as the standard to calculate the cellular antioxidant activity value, while 2′,7′-Dichlorofluorescin diacetate (DCFH-DA) was used as the fluorescence probe. ABAP was used as the free radical donor. PBS wash and no PBS wash treatments were used in this assay. Fluorescence intensity was measured at the excitation of 485 nm and emission of 535 nm for a dynamic fluorescein intensity analysis by the Multi-mode microplate reader (Molecular Devices, Sunnyvale, CA, USA). CAA value was calculated from the integrated area under the fluorescence versus time curve, and the results were expressed as micromole of quercetin equivalents (QE) per 100 g in fresh weight (μmol QE/100 g FW) for three replicates.

### 2.7. Statistical Analysis

Statistical analyses were performed using SigmaPlot software 12.3 (Systat Software, Inc., Chicago, IL, USA). The significance of relationships was calculated by the multivariate method. Data were analyzed among groups using one-way analysis of variance (ANOVA) and Duncan’s multiple comparison post-test using SPSS software 18.0 (SPSS Inc., Chicago, IL, USA). P-values less than 0.05 were regarded as statistically significant. All data were reported as mean ± SD of triplicate analyses.

## 3. Results

### 3.1. Titratable Acid and Total Sugar Contents in Sweet and Sour Wampee Fruits

Five sweet wampee (THP, ZFHP, CCTP, LTDH, and TXTP) and five sour wampee varieties (M3H, JZP, M4H, TXSP, and JXHP) were selected for analysis. According to Table 1, for the five sweet wampee varieties, the titratable acid contents were all below 0.100 g/100 g FW, and the total sugar contents were all above 10.00 g/100 g FW; meanwhile, the TS/TA ratios were all higher than 200, as compared to sour wampee samples. The variety of ZFHP had the lowest titratable acid content of 0.020 ± 0.002 g/100 g FW and the highest TS/TA ratio up to 817. For the five sour wampee varieties, the titratable acid contents were all above 0.700 g/100 g FW, and total sugar contents were between 8.00 g/100 g FW and 13.00 g/100 g FW; meanwhile, the TS/TA ratios were all lower than 20. JXHP had the lowest total sugar content of 8.810 ± 0.490 g/100 g FW, while M4H had the highest titratable acid content of 1.220 ± 0.030 g/100 g FW and the lowest TS/TA ratio of 9.94 among all the wampee samples. 

The average titratable acid content, total sugar content, and TS/TA ratio of five sweet wampees were 0.045 g/100 g FW, 13.93 g/100 g FW, and 395.9, respectively, while those of the five sour wampees were 0.895 g/100 g FW, 11.25 g/100 g FW, and 13.07, respectively. The results showed that the average titratable acid content of the sour wampee samples was 19 times higher than those of the sweet wampee samples. On the contrary, the average TS/TA ratio of the sweet wampee samples was 29 times higher than those of the sour wampee samples. However, the average total sugar content between sweet and sour wampee fruits did not show major difference, as did that of titratable acid and TS/TA ratio.

### 3.2. Total Phenolic Content in Sweet and Sour Wampee Fruits

The total phenolic content of ten different wampee varieties varied greatly from 49.25 ± 0.08 mg GAE/100 g FW (THP) to 829.1 ± 1.6 mg GAE/100 g FW (JXHP), according to Figure 2. The total phenolic contents of five sweet wampee varieties were all under 90.0 mg GAE/100 g FW, and the average content was 73.80 mg GAE/100 g FW. The lowest one was THP, which showed significant differences (*p* < 0.05) between the other four sweet wampee varieties. However, the total phenolic contents of five sour wampee varieties were all above 300.0 mg GAE/100 g FW, and there were significant differences between each other. The average total phenolic content of five sour wampee fruits was 521.7 mg GAE/100 g FW, which was six times higher than those of sweet wampee fruits. The results showed that the total phenolic contents in sour wampee fruits were obviously higher than sweet wampee fruits, which was a similar variation to the titratable acid contents in sour and sweet wampee fruits.

### 3.3. Total Flavonoid Contents in Sweet and Sour Wampee Fruits

According to Figure 3, THP contained the lowest value of total flavonoid content (54.41 ± 1.41 mg CE/100 g FW) among the ten different varieties, while JXHP had the highest value of 909.9 ± 177.9 mg CE/100 g FW, which was 16 times higher than that of THP. The total flavonoid contents of the five sweet wampee varieties were all lower than 150.0 mg CE/100 g FW, and there was no significant difference (*p* < 0.05) between each other. The average total flavonoid value of the five sweet wampee fruits was 99.82 mg CE/100 g FW. For the five sour wampee varieties, the total flavonoid contents showed major differences, and they could be classified into three levels. The low level was M3H with 281.3 ± 17.8 mg CE/100 g FW, and the medium levels were JZP, M4H, and TXSP, with the values of 392.5 ± 22.40 mg CE/100 g FW, 387.2 ± 22.5 mg CE/100 g FW, and 433.7 ± 13.4 mg CE/100 g FW, respectively. In addition, JXHP had the highest flavonoid content, which was three times higher than that of M3H. The average total flavonoid content of the five sour wampee varieties was 480.9 mg CE/100 g FW, which was four times higher than those of the sweet wampee varieties.

### 3.4. Phytochemical Profiles in Sweet and Sour Wampee Fruits

As shown in Table 2, eight phytochemical compounds were detected in sour wampee varieties, including syringin, rutin, benzoic acid, 2-Methoxycinnamic acid, kaempferol, hesperetin, nobiletin, and tangeretin, while just four of them were detected in sweet wampee varieties, including rutin, hesperetin, nobiletin, and tangeretin, and the other four were not detected (Appendix A). Syringin was the only one that was detected in all the sour wampee varieties and not detected in all the sweet wampee varieties. Rutin showed the highest level among the eight components in all the wampee varieties, but there was no difference between sweet and sour wampee varieties.

### 3.5. Total Antioxidant Activities of Sweet and Sour Wampee Fruits

The total antioxidant activities of ten different wampee varieties were determined by the ORAC assay. According to Figure 4, the ORAC value varied from 1.080 ± 0.140 mmol TE/100 g FW (THP) to 8.230 ± 0.920 mmol TE/100 g FW (JXHP). For the five sweet wampee varieties, CCTP showed the highest ORAC value of 2.020 ± 0.390 mmol TE/100 g FW and had significant differences (*p* < 0.05) with the lowest one, THP. The other three sweet wampee varieties (ZFHP, LTDH, and TXTP) did not show any significant differences between each other, either with CCTP or THP. The average ORAC value of the five sweet wampee varieties was 1.540 mmol TE/100 g FW. In the case of the five sour wampee varieties, the ORAC value could be divided into three levels: the high level of JXHP (8.230 ± 0.920 mmol TE/ 100 g FW), the medium levels of M3H (4.390 ± 0.350 mmol TE/100 g FW) and JZP (4.690 ± 0.640 mmol TE/100 g FW), and the low levels of M4H (3.380 ± 0.230 mmol TE/100 g FW) and TXSP (2.920 ± 0.470 mmol TE/100 g FW). There were significant differences (*p* < 0.05) between these three different levels. The average ORAC value of the five sour wampee varieties was 4.72 mmol TE/100 g FW, which was two times higher than those of the sweet wampee varieties. 

### 3.6. Cellular Antioxidant Activities of Sweet and Sour Wampee Fruits

The intracellular and cellular antioxidant activities were evaluated using the CAA assay, with PBS wash and no PBS wash methods. According to Figure 5, it was obvious that the CAA values of both the PBS wash and no wash methods in the sour wampee varieties were much higher than the sweet wampee varieties, especially of the no PBS wash samples. The CAA values in the no PBS wash samples of the five sour wampee varieties varied from 292.7 ± 28.5 μmol QE/100 g FW to 366.6 ± 39.6 μmol QE/100 g FW, and the average value of them was 335.9 ± 27.6 μmol QE/100 g FW, which was almost 15 times more than the average level of the five sweet wampee varieties. The CAA values in the PBS wash samples of the five sour wampee varieties varied from 67.95 ± 8.57 μmol QE/100 g FW to 188.3 ± 11.3 μmol QE/100 g FW, and the average value of them was almost 12 times higher than the average level of the five sweet wampee varieties.

### 3.7. Correlation Analysis

As shown in Figure 6, TPC, TFC, ORAC, CAA-no wash, and CAA wash values all showed significant positive correlations with TA, especially TPC and CAA. The TPC and CAA-no wash values showed significant negative correlations with TS/TA. In addition, there was a significant strong positive correlation between TFC and TPC (r = 0.972, *p* < 0.01). The ORAC value positively correlated with TPC (r = 0.902, *p* < 0.01) and TFC (r = 0.945, *p* < 0.01). The CAA-no wash value positively correlated with the TPC, TFC, and ORAC values. The correlation analysis suggested that the titratable acid could be an important factor that affected the contents of phenolics and flavonoids and the antioxidant activities of wampee fruits; furthermore, the total phenolic and flavonoid contents were major contributors to the antioxidant activities of wampee fruits. For the phenolic components, the correlation analysis showed that the syringin content was positively correlated with the TPC, TFC, and CAA values and that the benzoic acid content was positively correlated with the TPC, TFC, and ORAC values.

## 4. Discussion

In this study, the five sweet wampee varieties, with an average TS/TA ratio 29 times higher and an average titratable acid content 19 times lower than the five sour wampee varieties, showed significantly much lower levels of total phenolics, flavonoids, and antioxidant activities than the sour wampee varieties. Thus, the hypothesis was if there were any relationship between TS/TA and TA with TPC, TFC, and antioxidant activities. The correlation analysis showed significant positive correlations between TA with TPC, TFC, ORAC, and CAA values, while it showed significant negative correlations between TS/TA with TPC and CAANW values. These results suggested that the content of titratable acid in wampee fruit might have some relationship with the content of phenolics and flavonoids. In previous studies about fruit phenolics, flavonoids, and antioxidant activities, there was little focus on the relationship between titratable acid and these phytochemicals and antioxidant activities. In the research of the phenolic metabolism of the red currant through fruit ripening, the authors found that the phenolic contents slightly decreased in all three cultivars, while the sugar/acid ratios increased during development, and the average total phenolic values of four development stages of three cultivars were positively correlated with the average values of total organic acids [29]. Kim et al. analyzed the fruit quality and phenolic contents of 45 commercial cultivars of blueberry fruits grown in Korea [30], and Yilmaz et al. detected the total phenolic, acidity, and sugar contents of 31 edible wild pear fruits [31], while the relationship between total phenolics and acid and/or sugar were not analyzed in both papers. According to the data in the two papers about blueberry and pear fruits, the content of total phenolics and total acidity did not show a positive relationship, which was not consistent with the results of the wampee fruit.

As TA and TS/TA both significantly correlated with TPC and CAA values in this study, the question is if there were any phytochemicals correlated with TA or TS/TA? According to the correlation analysis in this study, among the eight chemicals detected from wampee fruits, only rutin (the one with the most content) showed a significant negative correlation with TS/TA (r = −0.671) but no significant correlation with TA; the others showed no significant correlation with TA or TS/TA. This may explain the relationship between TS/TA and total phenolics. However, if there is any relation between TA and TPC and TFC, or if they are just two parallels with no intersection, should be analyzed in further studies.

Phytochemical profiles of different wampee varieties in this study showed that sweet wampee varieties contain less phenolics and flavonoids because there were four more phytochemical compounds detected in sour wampee varieties than sweet wampee varieties. The four undetected chemicals in sweet wampee varieties of syringin, benzoic acid, 2-methoxycinnamic acid, and kaempferol may be the main factors that caused the differences of TPC and TFC between sweet and sour wampee varieties. The correlation analysis showed that syringin positively correlated with TPC, TFC, and CAA values (r > 0.6); benzoic acid positively correlated with TPC, TFC and ORAC values (r > 0.7); and kaempferol positively correlated with ORAC and CAA values (r = 0.83 and 0.64). However, benzoic acid and kaempferol were only detected in two or three sour wampee varieties, while syringin was detected in all the five sour wampee varieties but not in all the sweet wampee varieties. This suggested that syringin might be the key factor that caused the differentiation of TPC and TFC between the sweet and sour wampee varieties.

## 5. Conclusions

Sour wampee varieties analyzed in this study showed more total phenolic and flavonoid contents and higher antioxidant activity and cellular antioxidant activity than sweet wampees. The titratable acid content in the sour wampee fruits was significantly higher than in the sweet wampee fruits, while the total sugar content between the sour and sweet wampee varieties did not show significant differences. The titratable acid content may be the main factor that causes the taste difference between sour and sweet wampee fruits.

The correlation analysis showed significant positive correlations between TA with TPC, TFC, ORAC, and CAA values, which suggested that the content of titratable acid in wampee fruit might have some relationship with the content of phenolics and flavonoids. This study will provide an insight into the relationship between sugars, acids, and phenolic compounds in wampee fruit. Sour wampee varieties with high phenolics and antioxidant activity should be paid much attention by breeders to cultivate germplasms with high health care efficacy.

## Figures and Tables

**Figure 1 foods-11-01230-f001:**
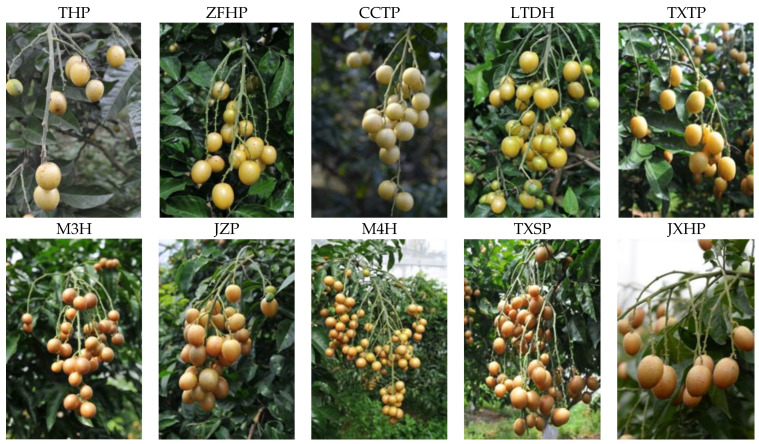
The varieties of wampee fruits. THP: TianHuangPi; ZFHP: ZaoFengHuangPi; CCTP: CongChengTianPi; LTDH: LuTianDuHe; TXTP: TaXiaTianPi; M3H: Min3Hao; JZP: JiZiPi; M4H: Min4Hao; TXSP: TaXiaSuanPi; JXHP: JiXinHuangPi.

**Figure 2 foods-11-01230-f002:**
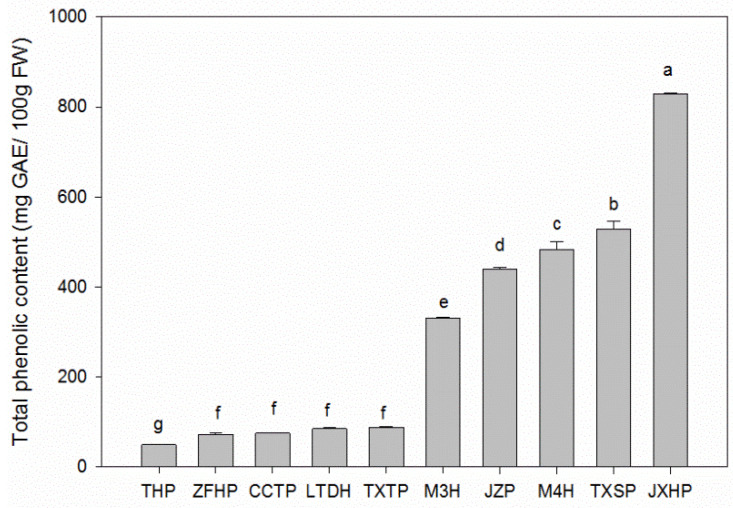
Total phenolic contents of five sweet and five sour wampee varieties. Bars with no letters in common are significantly different (*p* < 0.05).

**Figure 3 foods-11-01230-f003:**
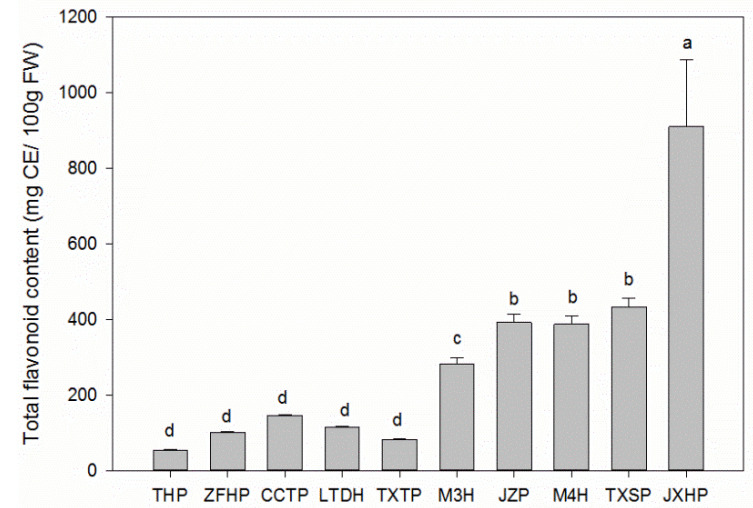
Total flavonoid contents of five sweet and five sour wampee varieties. Bars with no letters in common are significantly different (*p* < 0.05).

**Figure 4 foods-11-01230-f004:**
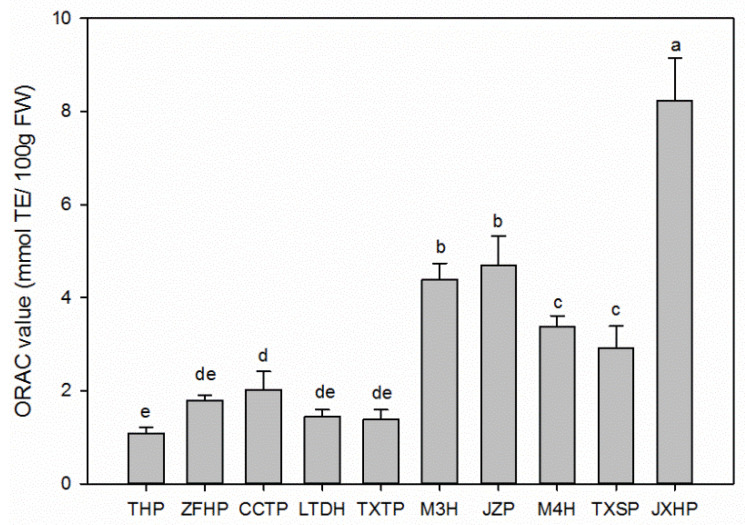
Total antioxidant activities of the five sweet and five sour wampee varieties obtained by the ORAC assay. Bars with no letters in common are significantly different (*p* < 0.05).

**Figure 5 foods-11-01230-f005:**
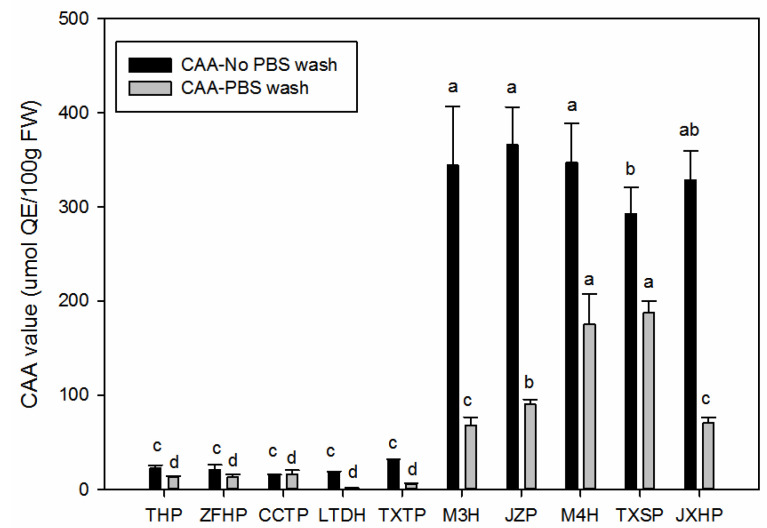
The cellular activities of the five sweet and five sour wampee varieties obtained by the ORAC assay. Bars with no letters in common are significantly different (*p* < 0.05).

**Figure 6 foods-11-01230-f006:**
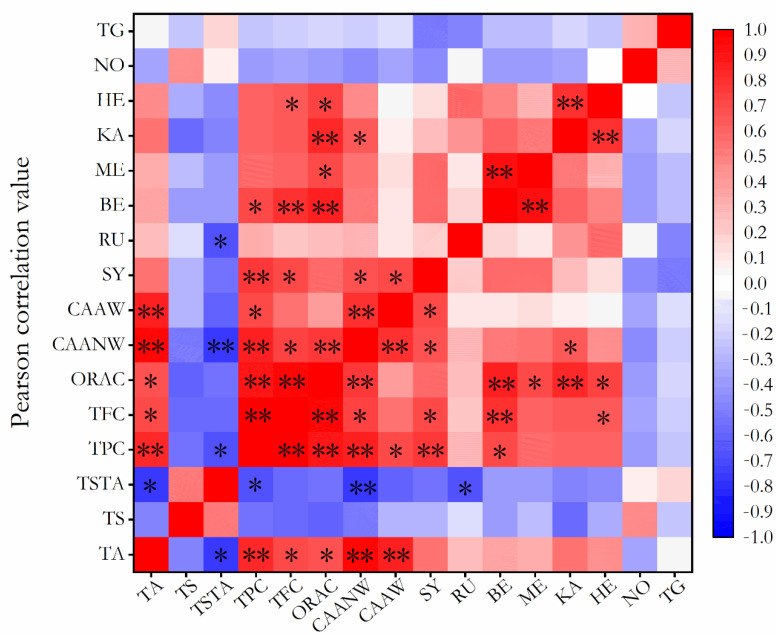
Heat map of the correlation analysis among titratable acid, total sugar, total phenolics, total flavonoids, phytochemical components, and antioxidant activities. TA: titratable acid; TS: total sugar; TSTA: total sugar/titratable acid; TPC: total phenolic content; TFC: total flavonoid content; ORAC: the oxygen radical absorbance capacity; CAANW: CAA value without PBS wash; CAAW: CAA value with PBS wash; SY: syringin; RU: rutin; BE: benzoic acid; ME: 2-methoxycinnamic acid; KA: kaempferol; HE: hesperetin; NO: nobiletin; TG: tangeretin. * *p* < 0.05, ** *p* < 0.01.

**Table 1 foods-11-01230-t001:** Titratable acid and total sugar contents of sweet and sour wampee varieties.

	Variety	Titratable Acid (TA) (g/100 g FW)	Total Sugar (TS) (g/100 g FW)	TS/TA
Sweet wampee	THP	0.047 ± 0.002 fg	10.42 ± 0.38 ef	222.0 ± 14.1 c
	ZFHP	0.020 ± 0.002 g	15.99 ± 0.20 b	817.1 ± 75.7 a
	CCTP	0.027 ± 0.005 g	11.65 ± 0.67 de	438.6 ± 109.4 b
	LTDH	0.049 ± 0.001 fg	13.90 ± 0.68 c	283.8 ± 6.4 c
	TXTP	0.081 ± 0.003 f	17.71 ± 0.73 a	218.1 ± 15.9 c
Sour wampee	M3H	0.927 ± 0.010 b	10.16 ± 1.71 f	10.97 ± 1.89 d
	JZP	0.719 ± 0.060 e	12.84 ± 0.39 cd	17.92 ± 1.17 d
	M4H	1.220 ± 0.030 a	12.12 ± 0.22 d	9.940 ± 0.150 d
	TXSP	0.770 ± 0.010 d	12.33 ± 0.87 d	16.01 ± 1.31 d
	JXHP	0.839 ± 0.020 c	8.810 ± 0.490 g	10.50 ± 0.35 d

Values with no letter in common in each column are significantly different (*p* < 0.05).

**Table 2 foods-11-01230-t002:** Phytochemical components of the sweet and sour wampee varieties (μg/g FW).

	Syringin	Rutin	Benzoic acid	2-Methoxycinnamic acid	Kaempferol	Hesperetin	Nobiletin	Tangeretin
THP	ND	63.54 ± 1.59 c	ND	ND	ND	1.060 ± 0.010 h	3.120 ± 0.080 g	6.570 ± 0.020 e
ZFHP	ND	41.25 ± 0.85 g	ND	ND	ND	1.110 ± 0.020 g	2.600 ± 0.02 i	5.620 ± 0.040 f
CCTP	ND	38.41 ± 0.36 g	ND	ND	ND	1.050 ± 0.010 g	9.520 ± 0.060 b	13.79 ± 0.02 c
LTDH	ND	59.01 ± 0.19 de	ND	ND	ND	1.150 ± 0.010 f	8.890 ± 0.060 c	14.87 ± 0.09 a
TXTP	ND	70.97 ± 5.74 a	ND	ND	ND	1.580 ± 0.010 c	11.78 ± 0.08 a	ND
M3H	0.11 ± 0.01 c	68.14 ± 0.43 ab	ND	ND	2.290 ± 0.030 a	1.790 ± 0.010 b	4.440 ± 0.320 e	6.880 ± 0.020 d
JZP	1.05 ± 0.09 b	55.79 ± 4.78 ef	27.87 ± 1.49 b	4.370 ± 0.290 a	1.090 ± 0.040 c	1.180 ± 0.030 e	2.810 ± 0.170 h	3.840 ± 0.420 h
M4H	0.24 ± 0.03 c	53.04 ± 0.95 f	ND	ND	ND	1.330 ± 0.010 d	6.500 ± 0.050 d	14.21 ± 0.10 b
TXSP	1.53 ± 0.01 a	61.84 ± 7.17 cd	ND	ND	ND	1.100 ± 0.020 g	3.740 ± 0.240 f	ND
JXHP	0.98 ± 0.08 b	65.68 ± 0.72 bc	35.01 ± 1.51 a	2.910 ± 0.150 b	2.070 ± 0.001 b	2.090 ± 0.010 a	3.910 ± 0.020 f	4.820 ± 0.080 g

ND: not detected; values with no letter in common in each column are significantly different (*p* < 0.05).

## Data Availability

The data presented in this study are available on request from the corresponding author.

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
