# Peer review of "Comparative Analysis of Phytochemical Profiles and Antioxidant Activities between Sweet and Sour Wampee (Clausena lansium) Fruits"

_foods, 2022, doi:10.3390/foods11091230_

Round 1

Reviewer 1 Report

This manuscript highlights a comparative study between the phytochemical profile and antioxidant activity of sweet and sour wampee fruits. 

Considering that wampee (Clausena lansium) is a highly consumed fruit in China, the study provides useful information regarding its chemical composition, titrable acidity, total sugars and antioxidant activity. 

The manuscript is well written, however, the Introduction part is brief and lacks references. Please put separate references for the medical effects of the functional compounds like: "cough [x], asthma [x, y], viral hepatitis [x] and so on. Putting 3-4 references together is not professional. 

Figures 2 to 6 are a bit large and throughout the text, the figure number and text do not correspond starting with figure 3 in the text which is actually figure 4.

The discussions are brief and are not backed up by similar research papers (one reference is not enough). Also, there are no conclusions presented. Please write some conclusions and add more references to improve the scientific quality of the manuscript. 

The content of the paper fits well with the journal’s scope after major revision. 

Author Response

The manuscript is well written, however, the Introduction part is brief and lacks references. Please put separate references for the medical effects of the functional compounds like: "cough [x], asthma [x, y], viral hepatitis [x] and so on. Putting 3-4 references together is not professional.

Response: Thank you for your advice. Introduction part has been supplemented with modifications and added with relevant references.

Figures 2 to 6 are a bit large and throughout the text, the figure number and text do not correspond starting with figure 3 in the text which is actually figure 4.

Response: Thank you for your advice. We are sorry for those mistakes. The figure size of Fig. 2 to Fig.6 was changed smaller than before. The figure numbers were all revised in the text from “figure 3” to “figure 6”.

The discussions are brief and are not backed up by similar research papers (one reference is not enough). Also, there are no conclusions presented. Please write some conclusions and add more references to improve the scientific quality of the manuscript.

Response: Thank you for your advice. The discussion part was modified with more relevant references. The conclusion part was added in the manuscript.

Reviewer 2 Report

The manuscript entitled Comparative Analysis of Phytochemical Profiles and Antioxi- 2 dant Activities between Sweet and Sour Wampee (Clausena 3 lansium) Fruits, is an interesting work especially it deals with compounds of natural origin (Plants), but for the publication in important journal like ''FOODS'', it is necessary to increase the quality by minor corrections, and among these corrections it has been mentioned below :

Abstract

Line 29: Keywords must be the first letter capitalized, e.g. 'wampee fruit' must be 'Wampee fruit

Introduction

Line 55 to 57: I think they discussed total polyphenols?

Line 57 CCTHP and THP, if possible you can add some lines to define.

Material and method

Line 77: correct "figure 1" to Fig.1

Line 90: correct "400 ml" to 400 mL

Line 93: why did you store the extracts at -20°C?

Line 109: correct "1.0mL/min" to 1 mL/min

Section 2.6 (line 119): why didn't you do or test the antioxidant activity of your extracts by the DPPH method, especially DPPH free radicals are the most known in this kind of test?

Statistical analysis

The manuscript has been well analysed, and the choice of statistical tests has been well chosen, especially the multivariate ones, but it is preferable to start first with the analysis of normality test by the "Shapiro-walks" test followed by the homogeneity test by the T test.

Results

Line 167: correct "Table 1" to "Table 1.

Table 1: Values should be 3 digit decimal places e.g. "10.42" should be 10.420 to comply with statistical analysis.

Section 3.3: values should be in dicimal places, please keep the decimal places the same e.g. 281.3 ± 17.76 should be 281.30 ± 17.76 and 387.2 ± 22.52 should be 387.20 ± 22.52.

Discussion

This section has been well structured and well presented with details, and also it has been well discussed with a clear and simple comparison.

Conclusion: Why was the conclusion missing in this work? It should be added. It is an important section for scientific researchers.

References: I think that there is a lack of references but that the work has been well respected.

Minor comment, I want to ask to make a small English correction of this manuscript.

Author Response

Abstract

Line 29: Keywords must be the first letter capitalized, e.g. 'wampee fruit' must be 'Wampee fruit.

Response: Thank you for your advice. We are sorry for those mistakes. The first letters of the Keywords were all capitalized in manuscript.

Introduction

Line 55 to 57: I think they discussed total polyphenols?

Response: Thank you. The “phytochemicals” in the sentence of line 55 to 57 was changed to “polyphenols”.

Line 57 CCTHP and THP, if possible you can add some lines to define.

Response: Thank you for your advice. Sentence of “with high soluble solids content and little content of titratable acid” was added to define “CCTHP and THP” in line 57.

Material and method

Line 77: correct "figure 1" to Fig.1

Response: Thank you. “figure 1” in line 77 was corrected to “Fig. 1”.

Line 90: correct "400 ml" to 400 mL

Response: Thank you. “400 ml” in line90 was corrected to “400 mL”.

Line 93: why did you store the extracts at -20°C?

Response: All the extracts were used for phytochemical composition determination and antioxidant activities analysis. The phytochemicals are stable at -20°C, and also the extracts were used for analysis during three months. The store condition would not affect the phytochemical contents and antioxidant activities in this study.

Line 109: correct "1.0mL/min" to 1 mL/min

Response: Thank you. “1.0mL/min” in line 109 was corrected to “1 mL/min”.

Section 2.6 (line 119): why didn't you do or test the antioxidant activity of your extracts by the DPPH method, especially DPPH free radicals are the most known in this kind of test?

Response: Thank you very much for your comments. We used ORAC and CAA methods for total and cellular antioxidant activities. ORAC method is recommended by AOAC for foods antioxidant activity evaluation, it is more sensitive and accurate for antioxidants. And the results could be compared with many other foods. CAA method evaluated extracts antioxidant activity in cellular levels, it could reflect antioxidant activities in physiological status. That’s why we chose the two methods for antioxidant activity evaluation.

Statistical analysis

The manuscript has been well analyzed, and the choice of statistical tests has been well chosen, especially the multivariate ones, but it is preferable to start first with the analysis of normality test by the "Shapiro-walks" test followed by the homogeneity test by the T test.

Response: Thank you very much for your suggestion. Ten typical varieties of wampee fruits were selected for comparation and evaluation in this study, the data would be more suitable for T test analysis. We hope to extend more characters and genotypes in the future, and then the data would be used for "Shapiro-walks" test.

Results

Table 1: Values should be 3 digit decimal places e.g. "10.42" should be 10.420 to comply with statistical analysis.

Response: Thank you for advice. We have revised and uniformed all the data expression in the manuscript as 4 significant digits rules this study, which was common as international units. And the data showing would be more objective and scientific.

Section 3.3: values should be in dicimal places, please keep the decimal places the same e.g. 281.3 ± 17.76 should be 281.30 ± 17.76 and 387.2 ± 22.52 should be 387.20 ± 22.52.

Response: Thank you for your advice. It has been revised in the manuscript, and the data expression was uniformed as 4 significant digits in the whole manuscript.

Conclusion: Why was the conclusion missing in this work? It should be added. It is an important section for scientific researchers.

Response: Thank you for your advice. “Conclusion” part was added in the manuscript.

References: I think that there is a lack of references but that the work has been well respected.

Response: Thank you for your advice. The introduction and discussion parts were both modified with 19 more references added.

Minor comment, I want to ask to make a small English correction of this manuscript.

Response: Thank you for your advice. The manuscript has been deeply revised in English writing. We hope the revision suitable for acceptance.

Reviewer 3 Report

This article analyzed the phenolics, flavonoid contents and antioxidant activities of wampee fruits of more than one hundred and fifty different germplasms, and found that sweet wampee varieties contained less phenolics and flavonoids, and also showed lower antioxidant activities than those of sour wampee varieties. It will help the scientific community to develop novel food with enriched phytochemicals.  Before recommending this article for publication, there are some shortcomings for that should be resolve.

Abstract

This section is well written, but the author has focused too much on the results part. The authors should present summarized methods in the abstract section.

In addition, the sentences are very long which void the main concept of the sentences.

In the last section add one to two sentences of conclusion and future recommendation.   

Introduction

Introduction section is well written, but information is limited.

The authors are directed to add economic and medicinal importance of both the fruit.

Add significance of phytochemicals and its anti-oxidant mechanism by citing recent literature.

https://doi.org/10.1016/j.jep.2021.114515, https://doi.org/10.1016/j.chnaes.2021.03.009,  

In the last paragraph discuss aims and objectives of the study and novelty.     

Materials and Methods

Experiment is well designed, and methodology is well written.

Present a table containing details of the varieties would be better.  

The author must state that which conditions and precautions were followed in extract or sample preparation.

The information on sample preparation is not enough.

Results

“Bars with no letters in common are significantly different (p < 0.05)” The sentence must be change into Bars with same letters are significantly different at (p< 0.05).  

Discussion

Discussion is well justified however the authors should present and discuss some relevant studies with their results.

Conclusion is missing.

Author Response

Abstract

This section is well written, but the author has focused too much on the results part. The authors should present summarized methods in the abstract section.

Response: Thank you for your advice. Summarized methods was added in the “Abstract” part.

In addition, the sentences are very long which void the main concept of the sentences.

Response: Thank you for your advice. We have revised the manuscript as it more readability.

In the last section add one to two sentences of conclusion and future recommendation.

Response: Thank you for your advice. One sentece of conclusion and future recommendation was added in the “Abstract” part.

Introduction

Introduction section is well written, but information is limited.

The authors are directed to add economic and medicinal importance of both the fruit.Add significance of phytochemicals and its anti-oxidant mechanism by citing recent literature. https://doi.org/10.1016/j.jep.2021.114515,  https://doi.org/10.1016/j.chnaes.2021.03.009. In the last paragraph discuss aims and objectives of the study and novelty.     

Response: Thank you for your advice. More literatures about the medicinal importance of wampee were added in the “Introduction” part. The aims and novelty of this study was added in the last paragraph in the “Introduction” part.

Materials and Methods

Experiment is well designed, and methodology is well written.

Present a table containing details of the varieties would be better.

Response: Thank you for your advice. The details of samples were added in the manuscript.

The author must state that which conditions and precautions were followed in extract or sample preparation.

Response: Thanks for your advice. It was added in the manuscript. And also, all the experiments were strictly operated with previous reports in our lab or other publications, the more details of operation could be found according to corresponding references.

The information on sample preparation is not enough.

Response: Thank you for your advice. More information was added in “2.1 sample preparation” part.

Results

“Bars with no letters in common are significantly different (p < 0.05)” The sentence must be change into Bars with same letters are significantly different at (p< 0.05).

Response: Thanks for your comments. The significant analysis showed that different letters means significant differences at bars. Therefore, it should be better to express as “Bars with no letters in common are significantly different (p < 0.05)”.

Discussion

Discussion is well justified however the authors should present and discuss some relevant studies with their results.

Response: Thank you for your advice.The discussion part was modified with more relevant references.

Conclusion is missing.

Response: Thank you. The conclusion part has been added in the manuscript.

Round 2

Reviewer 1 Report

The manuscript was corrected according to the specified instructions and the Introduction has been supplemented with other relevant references. 

Figure size and numbers were corrected acordingly to instructions. 

Disscusions were modified and similar studies were added as references. 

The added conclusions improved the scientific value of the manuscript. 

I consider the article suitable for publication in the present form.